# Challenges and solutions: surveying researchers on what type of community engagement and involvement activities are feasible in low and middle income countries during the COVID-19 pandemic

Karolin Kroese,[1] Katie Porter ,[2] Heidi Surridge,[2] Doreen Tembo[2]

KK, KP, HS and DT contributed equally.

[1]NIHR Global Health Research Unit on Global Surgery, University of Birmingham, Birmingham, UK
[2]Wessex Institute, Faculty of Medicine, University of Southampton, Southampton, UK

**Correspondence to**
Katie Porter;
katie.porter@nihr.ac.uk

## ABSTRACT

**Objectives** Measures to limit the spread of infection during the COVID-19 global pandemic have made engaging and involving members of the community in global health research more challenging. This research aimed to explore how global health researchers adapted to the imposed pandemic measures in low and middle income countries (LMICs) and how they overcame challenges to effective community engagement and involvement (CEI).

**Design** A qualitative two-stage mixed-methods study involving an online survey and a virtual round table.

**Setting** The survey and round table were completed online.

**Participants** Of 53 participants, 43 were LMIC-based or UK-based global health researchers and/or CEI professionals, and 10 worked for the National Institute for Health Research or UK Government's Department of Health and Social Care.

**Outcome measures** This study aimed to capture data on: the number of CEI activities halted and adapted because of the COVID-19 pandemic; where CEI is possible; how it has been adapted; what the challenges and successes were; and the potential impact of adapted or halted CEI on global health research.

**Results** Pandemic control measures forced the majority of researchers to stop or amend their planned CEI activities. Most face-to-face CEI activities were replaced with remote methods, such as online communication. Virtual engagement enabled researchers to maintain already established relationships with community members, but was less effective when developing new relationships or addressing challenges around the inclusion of marginalised community groups.

**Conclusions** COVID-19 has highlighted the need for contingency planning and flexibility in CEI. The redesigning and adopting of remote methods has come with both advantages and disadvantages, and required new skills, access to technology, funding, reliable services and enthusiasm from stakeholders. The methods suggested have the potential to augment or substitute previously preferred CEI activities. The effectiveness and impact of these remote CEI activities need to be assessed.

### Strengths and limitations of this study

► This study used a two-stage mixed-methods evaluation using an online survey and an international round table to capture the impact of the COVID-19 pandemic on community engagement and involvement activities.

► The participants represent a broad variety of low and middle income countries across multiple continents, therefore providing a good overview of community engagement and involvement (CEI) activities across the board.

► The level of seniority and expertise of participants is varied, ensuring an accurate representation of CEI adaptation across the global health research network.

► Data were triangulated and coded to generate common emergent themes from the qualitative data to understand the main challenges and barriers research teams have faced, and how their CEI activities have been adapted in global health research.

► Due to the sample size and time frame when data were collected, this study only provides a snapshot of the CEI activities being done in global health research early on in the pandemic (June/July 2020), but it provides an opportunity for shared learning and contingency planning in a unique situation.

## INTRODUCTION

The development and expansion of the UK National Institute for Health Research (NIHR) and other UK-based funding bodies into the global health arena has necessitated a rethink of what is known as patient and public involvement (PPI) in the UK. UK-based global health researchers had to explore ways to involve community members in low and middle income countries (LMICs) where their research is taking place, using paradigms that are feasible,

accepted and effective in a local context. In global health research, the involvement of public members is internationally more commonly referred to as community engagement,[1] and within NIHR's global health research community as community engagement and involvement (CEI).

While there are some variations in the definition of CEI, the one used by the US Centers for Disease Control and Prevention summarises the essence of community engagement as: '…the process of working collaboratively with and through groups of people affiliated by geographic proximity, special interest, or similar situations to address issues affecting the well-being of those people'.[2] Community can broadly be defined as a minimum social unit who have a stake in the proposed research. This could include individuals, groups, organisations, government bodies, peer groups or social networks, and those who can influence or who may be affected by the research.[3] Community involvement can therefore be defined as the community actively participating or being involved in any aspect of the research cycle.[4]

Carrying out meaningful CEI within research is important as it reduces the potential for exploitation of communities and facilitates the implementation of health research.[4] Furthermore, involving communities in LMICs in health research funded by high-income countries (HICs) serves to address possible ethical concerns and cultural differences such as distrust and concerns about scientific colonialism.[5] If done properly, for example, following the UNICEF standards,[1] CEI can help ensure that benefits of research can be used by those in need, build on local capacity and support a true culture change in low-resource settings by empowering people at grassroots level.

As COVID-19 developed into a global pandemic and countries went into lockdown, engaging and involving members of the public in research became far more challenging. In this paper, we define 'lockdown' as large-scale physical distancing measures and movement restrictions,[6] however we are aware that this term is specific to a UK context and not necessarily used globally. Similarly, 'post-lockdown periods' are not a unified, global point in time, and are varied in both severity and time frames.

Solutions to involving the public in research during lockdown in the UK rapidly evolved around online communication platforms, however, this may not be feasible or poses additional challenges in LMICs. In LMICs, CEI has focused heavily on face-to-face communication due to limited access to technology and the internet. Traditionally, face-to-face communication has been relied on as a direct way to overcome barriers to involvement and is a cultural preference in some countries.[7]

This project focused on investigating how CEI in low-resource settings developed and adapted to the restrictions imposed as a response to the COVID-19 pandemic. We aimed to understand the impact the restrictions had on CEI in LMICs and the main challenges researchers had to overcome. We highlight how CEI is being redesigned

to address some of these barriers, explore the limitations with these solutions and discuss the implications for CEI.

## METHODS

We used a two-stage mixed-methods evaluation using an online survey and an international round table. Data were collected across June to early July 2020. Three of the authors work for the NIHR, but are not directly involved in the commissioning process, and so were able to use their networks to contact and involve NIHR-funded awardees and the NIHR CEI Global Health Advisory Network in this research.

### Survey

An online survey was designed to capture the impact of COVID-19 on CEI activities, the main challenges and barriers research teams have faced, and how CEI activities have been adapted (see online supplemental appendix 1). The survey captured quantitative data on demographics of participants and their research, as well as qualitative free text data on their CEI pandemic experiences. LMIC-based research team members, including public members and NIHR CEI professionals, reviewed the survey and contributed to its development.

The sampling used a combination of purposive and snowballing techniques.[8] The survey link, an invitation email and information sheet were sent to all NIHR-funded (and thus UK Aid-funded) global health researchers and infrastructure awardees (n=70). The survey invitation was also posted on the Mesh Community Engagement Network website to access non-NIHR-funded researchers. NIHR colleagues and networks shared the survey invitation with their own networks in the Global Health Research Community.

### Round table

A round table discussion exploring the same topic was hosted as part of this research project. Participants were invited from the NIHR's Global Health CEI Advisory Network,[9] with members based in the UK, as well as LMICs, all undertaking research with extensive CEI components. Participants therefore offered a wealth of experience and knowledge in undertaking CEI in different LMIC contexts. All members who took part had completed consent forms and received an information sheet prior to the round table. Three of the authors work regularly with this network to help shape and inform NIHR's CEI portfolio, and so relationships with participants were already established.

Two of the authors facilitated the online round table (via Zoom). The discussion was guided by a series of questions agreed by the research team prior to the round table (online supplemental appendix 2). Qualitative research notes from the round table were taken by the research team and circulated to participants for validation.

### Data analysis

Using thematic data analysis, three individual members of the team analysed the triangulated survey and round

table data. Collection and analysis of the quantitative data were supported by Microsoft Excel. NVivo software supported the analysis of the qualitative data. Themes were discussed by the research team and went through several iterations as discussions progressed. The resultant common emergent themes are described in the Results section.

## Patient and public involvement

As this research aimed to capture the researchers' perspective, with the primary target audience being global health researchers and funders, patients and community members were not specifically asked to be involved in the design and undertaking of this research. However, as part of the planning process, representative individuals from LMIC-based research teams, including members of the public and CEI professionals, were asked to review the questions to ensure relevant information on CEI activities was being captured. They were also asked to trial the online survey tool for usability and clarity prior to it being circulated.

Furthermore, when participating in the survey, lead researchers were encouraged in the invitation email to consult with their wider team prior to submitting their response. This included community members, where applicable and relevant. The collected data describe how research teams are engaging and involving community members in the research process.

This was a reactive piece of monitoring and evaluative work initiated by NIHR and an initial query raised by a global health CEI specialist (KK) with a primary purpose of understanding the impact of the pandemic on the research community working in LMICs. This in turn forms learning for the funder on changing needs of researchers.

## RESULTS
### Survey participant demographics

Thirty-one (31) surveys were returned. Information on participant demographics can be seen in table 1. Of

| Table 1 | Participant demographics | |
|---|---|---|
| | **Survey participants** | **Round table participants** |
| Total number of participants | 31 (100%) | 22 (100%) |
| Where are they based? | | |
| UK | 22 (70%) | 15 (68%) |
| Central America | 1 (3%) | |
| North Africa | 1 (3%) | |
| East Africa | 2 (6%) | 3 (14%) |
| West Africa | 1 (3%) | |
| South Africa | 1 (3%) | 1 (5%) |
| South East Asia | | 2 (9%) |
| South Asia | 2 (6%) | 1 (5%) |
| Not given | 1 (3%) | |
| Role | | |
| CEI lead/manager | 3 (10%) | 2 (9%) |
| Project/programme manager | 7 (23%) | |
| Professor or associate professor | 8 (26%) | 3 (14%) |
| Research fellow | 1 (3%) | |
| Consultant | 1 (3%) | 3 (14%) |
| Medical worker or officer | 2 (6%) | |
| Research assistant | 2 (6%) | |
| Research associate | 2 (6%) | |
| Researcher | | 2 (9%) |
| PhD student | 1 (3%) | |
| Lecturer | 1 (3%) | 1 (5%) |
| Chief Executive Officer | 1 (3%) | |
| Director | 1 (3%) | 1 (5%) |
| NIHR/DHSC staff member | | 10 (45%) |

CEI, community engagement and involvement; DHSC, Department of Health and Social Care; NIHR, National Institute for Health Research.

**Table 2** Unaffected, paused and amended CEI activities of survey participants

| Participants with ongoing or partly ongoing research studies | 28 (90% of survey participants) | No changes to any or most CEI activities | 4 (14% of participants with ongoing research studies) |
| --- | --- | --- | --- |
| | | Paused some or all CEI activities | 24 (86% of participants with ongoing research studies) |
| | | Amended some or all CEI activities | 22 (79% of participants with ongoing research studies) |

Please note there is overlap between the amended and paused responses.
CEI, community engagement and involvement.

these 31, 27 (87%) were in receipt of NIHR funding, but many reported support from other funders (see online supplemental appendix 3). Most participants reported on multiple research projects in different LMICs, in thematically diverse research areas. Participants' research was being conducted mainly on the African continent (61 out of 116 individual projects for this question, 53%), followed by Asia (39 individual projects, 33%) and South America (10 individual projects, 9%).

### Round table participant demographics
Twenty-two (22) people took part in the round table: 7 (32%) were LMIC-based global health researchers, 5 (23%) UK-based global health researchers, and 10 (45%) UK-based NIHR and Department of Health and Social Care staff members (see table 1 for more information).

### Impact of COVID-19 restrictions on CEI activities in LMIC-based research
All respondents stated that initially all their studies included CEI components, which ranged from consultative activities like community advisory boards/groups and focus groups to more involved activities such as co-production of study materials, or members of communities undertaking data collection.

Due to COVID-19, overall, 83% of survey participants' planned CEI activities had to be amended or paused. Details on this can be found in tables 2 and 3.

### Reasons for halting CEI activities
#### Restriction of face-to-face interaction
During country-specific periods of lockdown, which included non-essential travel restrictions and social distancing measures, all face-to-face CEI activities had to be paused. Restrictions meant that researchers were unable to travel to different geographical areas, especially

**Table 3** CEI activities being planned

| Participants currently planning new LMIC-based research studies | 14 (45% of all survey participants) |
| --- | --- |
| New studies containing CEI components | 11 (78% of all new studies) |

CEI, community engagement and involvement; LMIC, low and middle income country.

remote locations, for research purposes, such as providing interventions or collecting data.

> CEI which involved movement of researchers from one area to another […] were also halted to avoid the spread of COVID-19. This has had a serious negative impact on CEI […] the expected beneficiaries did not benefit anything from the CEI community dementia awareness raising planned activities. (Survey respondent 25)

Local engagement activities throughout the research process were reported by many as involving regular face-to-face communication with researchers, multilevel government and organisational stakeholders, as well as community members. In Africa, many engagement activities are carried out face-to-face with community leaders and other representative bodies, as a culturally well-respected form of communication (*round table member 2*).

> We had planned to develop all materials through co-design and to undertake a series of focus groups for this purpose within India. However, we have been unable to complete this. (Survey respondent 4)

> The team in Nepal were already heavily involved in a number of community awareness campaigns, using radio, tv, local schools, women's cooperatives etc. Due to Covid-19 these were all cancelled. (Survey respondent 3)

#### Diversion of resources
In some countries, governments decided that all non-essential research should be halted. Others found that their research partners were diverted to other pandemic-related activities or research.

> In two of the three countries the ethical and regulatory authorities have ordered that all 'non-essential' research activities are stopped. We have been able to continue our interventional clinical trials as they are deemed to be of direct importance to patient care, but the CEI activities are viewed as non-essential. (Survey respondent 23)

> Due to the pandemic, and our respiratory specialty, our teams have been pulled to the front-line—while in key stages of completing our programme's milestones! (Survey respondent 7)

## CEI solutions to pandemic measures
### Remote methods of CEI

Due to pandemic restrictions, online methods replaced planned face-to-face activities. These have been highlighted as positive alternatives for one-way communication, such as sharing documents or videos for training and awareness raising. A key benefit is that resources can be shared widely and repeatedly with minimal effort, cost and time.

> We now have a video recording of all our training and don't hand out paper manuals… Everyone receives a copy of the training on a USB and is free to revisit it whenever they need to. (Survey respondent 10)

Researchers in the Occupied Palestinian Territories found that large-scale online campaigns to raise awareness of burns studies have proven effective, but where materials could not be distributed online, they planned to combine their CEI activities, in this case health research awareness raising materials, with ongoing COVID-19 initiatives.

> Pamphlets and materials were prepared for a house-to-house community awareness campaign before COVID-19 lockdown. They [Ethiopia research team] are working with the Ministry of Health to see if they can combine their work on spreading COVID-19 messages which will be house-to-house […] with burn prevention messages. (Survey respondent 3)

For two-way communication, researchers quickly adapted to using video calls (Zoom, Skype, GoToMeeting), phone, text, WhatsApp, WeChat and similar apps. Online workshops and questionnaires were also introduced as participants adapted to the restrictions. Researchers pointed out that WhatsApp groups and Zoom meetings are helpful communication tools across borders and for communication between meetings. Some are keen to maintain communication via these platforms after country-specific pandemic restrictions are lifted, in addition to face-to-face meetings, to ensure continuity of contact with community stakeholders.

> As the groups have been engaged from the initial stages of the research they are keen to continue and provide input into the developing projects […] WhatsApp groups are the most useful method of communication for the CEI groups in the countries in which we work. The groups used this method between meetings, but also complete group meetings now using this platform. (Survey respondent 15)

> We have been pleasantly surprised by the relatively good internet access available to our researchers in Nepal who have been working from home. This is encouraging as it suggests that domestic internet access is better and more widespread than we initially imagined and may mean that some CEI activity is potentially feasible. (Survey respondent 17)

Online communication methods were well received by young people in some LMICs. For example, in Nepal, Facebook groups were created to help young men discuss their mental health and well-being during lockdown (*round table member 6*).

### Planning of post-lockdown CEI activities

The enforced period of restrictions allowed researchers to develop and adapt research and CEI plans for future post-pandemic research projects, while recognising that flexibility may be needed long term.

> All activity has been delayed due to COVID-19 but we have been able to do all the planning and will be ready to do face-to-face work once restrictions in Uganda and Tanzania are over. Things like filming and getting groups of patients together has not been possible due to restrictions within the country. (Survey respondent 5)

## Challenges with adapted CEI activities
### Small group meetings post-lockdown

In some countries, participants reported that small group gatherings were possible as restrictions began to ease. However, the extra precautions of personal protective equipment (PPE) and smaller community group meetings have higher resource implications for CEI.

> The inability to hold massive community meetings at halls during this time is really challenging […] Currently, we have broken down our scale to less than 30 per meeting which means more meetings and outings than preferred. (Survey respondent 10)

A research team also reported that on return to the community, post-lockdown, wearing the necessary face masks and PPE resulted in caution and mistrust as they were not easily identifiable (*round table member 7*).

### Community and stakeholder availability and priorities

Research partners in LMICs have found themselves diverted to the pandemic response, while others have reduced CEI in order to keep their research sustainable. Furthermore, the pandemic restrictions resulted in community members having to prioritise basic needs of food, work and care for their families over engaging with researchers.

> As our partners are all clinicians with a focus on respiratory care, many of them are being pulled into the front line. (Survey respondent 7)

> Fear from getting infected so they refuse participation in research. Lockdown leaves them with few hours to respond to researchers, their priority is their children, food etc. (Survey respondent 11)

### Inclusivity: access to, and connectivity with, remote technology

While remote means of communication have been the main CEI method during COVID-19, it has not been

without its challenges and limitations. The use of digital platforms relies on the individuals having access, funds and ability to use the technology and the internet.

In India, access to a smartphone was common but many could only access the internet on their devices at work, where free WiFi was provided (*survey respondent 4*). One research team working in sub-Saharan Africa noted that WhatsApp has enabled researchers to continue their CEI as even though computer access is rare, mobile phone access is common (*survey respondent 15*). Others working in Liberia and Nepal reported no or limited access to computer or mobile devices. Where devices were available, participants reported that larger scale remote meetings, workshops and training were problematic due to unreliable internet connections. To surmount some of these issues, respondents suggested that research teams could provide data allowances and technology for community stakeholders already involved with their research to ensure ongoing communication. Obviously, this has cost implications and may need to be costed into funding applications under CEI budget.

> The challenge of poor Internet connectivity often frustrated seamless virtual meetings with colleagues and stakeholders. We had to spend more funds purchasing alternative data service provider modems, and increased budget providing data for team members working from home. (Survey respondent 31)

> Communication is often challenging as a lot of the groups that we engage with have limited access to technology platforms—the digital divide. (Liberia; Survey respondent 27)

Teams who set up virtual groups using text-based platforms, such as WhatsApp, faced language and literacy barriers. One research group reported using voice notes as a way around this (*survey respondent 24*). Participants also noted that being able to see everyone's numbers on WhatsApp raised safeguarding concerns.

Remote communication was also thought to further marginalise some groups, for example, in a study in Pakistan, husbands were often gatekeepers to women's phone access (*survey respondent 12*). Another team working with religious minority groups and in slum settlements found that many people, especially women and older people, had no internet access (*round table member 7*).

> Unable to have face to face contact since lockdown due to COVID-19 so using telephone but this can be difficult for some groups such as women in Pakistan who may only have access to their husband's telephone[…] Could miss some of the most vulnerable people in the community by only having telephone communication. (Survey respondent 12)

Many study participants reported that online and digital communication for CEI activities highlighted the social and economic inequalities within and between countries.

### Quality of relationships built remotely

Some participants referred to remote communication as 'second best'—an interim measure with concerns based on quality. Telephone and virtual meeting conversations were described as shorter, formal and less natural than face-to-face interactions. They were deemed tolerable for established relationships, but not ideal for making crucial stakeholder connections and establishing new relationships. Respondents therefore tended to focus on maintaining and developing relationships that were established before lockdown.

> Face to face communication is highly valued in Nepali society. Taking the trouble to visit someone in person demonstrates that you value them and their opinion, and is essential for building relationships. (Nepal; Survey respondent 17)

> We have also used online Zoom webinars, which are useful in engaging people and groups, however, compared to face-to-face meetings, they lack the networking and 'connection' which face-to-face conferences and meetings have (…) this is very personal and valuable. (China; Survey respondent 7)

Some research teams have reverted to online surveys and posting information, where previously these engagements would have been conducted face-to-face and hence interactively.

## DISCUSSION
### Impact of COVID-19 on CEI

CEI activities are important for researchers in building and maintaining trust with local communities,[10] and so the authoritarian nature of strict lockdowns and lack of information on COVID-19 in some LMICs have put this in jeopardy. As lockdown restrictions ease globally, researchers need to consider elevated levels of suspicion, especially among new contacts. The use of PPE by researchers in particular was identified as an aspect that created mistrust within the community as researchers were not easily identifiable, however this mistrust may also stem from a lack of access to PPE for community members. If researchers have access to PPE and the community does not, this may create a power imbalance and potentially lead to feelings of mistrust.

A pandemic situation re-emphasises the community's priorities; basic needs of access to food, work, health and safety are prioritised over involvement with research.[11] As a way to keep communities engaged, researchers could explore options to partner with others delivering community health interventions, such as COVID-19 vaccination programmes. These grassroots community workers, if adequately resourced, may be able to deliver wider CEI activities alongside their own interventions. Funders also need to understand shifting community needs and priorities and potentially allow for reprioritisation in projects in such circumstances. Also as research continues, project

teams will need to consider the ethics of continuing CEI activities in relation to the needs of the community, especially as this pattern may continue as communities face prolonged impact of pandemic restrictions.

## Impact of COVID-19 on research

The quality and impact of the amended CEI activities and impact on the research they support have yet to be investigated, especially when considering the limitations to remote CEI activities. It has been reported that CEI adaptations to pandemic measures have ultimately been inequitable and the needs of resource-poor and underrepresented communities were given little consideration.[12] The rapid response of many COVID-19 research projects has caused some to abandon regular engagement processes and focus on the voice of 'professionals'.[13] In emergency pandemic situations, we need to ensure that the voice of the community and public should not be lost.[14]

Reduced or poorer quality CEI may affect the outcomes of a research study, its impact[1] and potentially opportunities for future funding. Some of the participants in this study paused CEI activities that were not feasible to do while restrictions were in place and continued with research activities that were. The effect of this has not yet been explored with the projects reported on here, however stopping CEI activities while moving other aspects of the research forward may imply that the community's role is not vital in the design and delivery of the research. It must be considered that CEI activities were paused as what was planned was no longer feasible and research teams were in the transition process of adapting their CEI and research activities to adhere to pandemic restrictions.

In periods where CEI activities must be paused, time can be used to plan and adapt future CEI activities. This time can also be used to develop the skills of the research team and community members, in particular online communication skills. Moving forward, there may also need to be contingency plans in place for managing CEI in areas with high prevalence of unpredictable communicable diseases. These contingency plans would need to include funding, access and support in the alternative means of communication and activities for all stakeholders involved in CEI. This is naturally an issue that reaches beyond CEI and research alone.

## Pros and cons of virtual engagement

Good working relationships with community stakeholders are essential to ensure the value and quality of research,[15 16] however trusted relationships take time to develop. Building and maintaining meaningful relationships with the community are an important component of CEI,[17] and this study has shown that pre-pandemic relationships may be sustained despite the lack of face-to-face engagement. Remote two-way communication methods were reported as useful in maintaining continuity of existing relationships during the pandemic and were already a well-established communication method.

Collaboration and co-production of research relies on the development of trusting relationships[6 10 17] and developing these relationships relies on regular, open communication with the community.[18 19] Virtual CEI methods have been less effective in reaching out and engaging new communities, especially isolated vulnerable groups.[20] A feasible solution to better engage relevant community stakeholders suggested by LMIC researchers in this study is to use grassroots organisations and mobilise local civil society groups, which has been an effective way during COVID-19 to reach communities.[21 22] Existing relationships can then be built upon which may also enable rapid response mechanisms, especially with marginalised communities.

## Inclusivity concerns
### The digital divide

Often, community members in LMICs have no or limited access to devices or the internet, and when they do, internet connection may be unstable or confined to certain areas. Thus, lockdown restrictions have meant that researchers struggled to reliably reach some community members. This means the most marginalised are even more likely to be excluded. This is an issue in HICs as well as LMICs, highlighting the need for non-technology-based engagement.[20] Although participants suggested that researchers can pay for data allowances and technology, this raises questions around whether funders would be willing to cover these costs. Generally, remote platforms for CEI activities have differed from what was originally planned, and the pandemic has required researchers to be creative. Research teams need to think flexibly about what is feasible, and how various technologies and online tools can be used in the longer term. The sudden onset and fluctuating incidence of this pandemic and subsequent restrictions means ongoing uncertainty and contingency plans for CEI should be in place. Like researchers, funding bodies could consider room for flexibility for CEI activities proposed by the applicant, for example, welcoming alternative and multiple scenario CEI activities for proposed research projects and allowing for reallocation of funds within the budget.

Some digital video software might provide a solution for people who have access to a phone, but no internet connection, as people can dial into meetings via phone calls. It could also be more cost-effective, accessible and increase reach and diversity.[23 24] However, for community members to use the application and feel confident to contribute in meetings, training might be required. Depending on the capacity of the research team, this may pose an issue when trying to involve large numbers of community members.

Age may also be a factor as young people, particularly those in developing countries, who have access to gadgets or the internet may already be comfortable using these methods of communication.[25–27] Thus, engagement with this age group via online platforms may be easier for researchers compared with engaging older age groups.[27]

Technology also raises concerns of data privacy in LMICs and HICs alike and is a concern in PPI/CEI, as well as telemedicine.[28] [29] However, data privacy is not a new concern especially in healthcare.[29] An adequate solution for data privacy is yet to be found.[28]

Close-knit communities often rely on 'word of mouth', where information is passed verbally, in-person to members of the community who do not have access to technology.[30] This approach should not be encouraged in pandemic situations and is prone to perpetuating inaccuracies through onward translation.

Research teams, as well as funders, will need to maximise their knowledge and creativity with virtual engagement while accepting that there may be limitations. Maintaining trust and relationships with community members is now more important than ever to ensure that they do not feel abandoned in times of crisis, to make research relevant and beneficial and to allow for involvement of key stakeholders when designing and implementing new studies remotely (for example, rapid response for COVID-19 studies). The use of digital tools is a way to do this, although perceived as not as inclusive or with the same depth as face-to-face communication. This may change with improving knowledge, familiarity and access. Remote communication also ensures ongoing transparency by keeping community stakeholders informed of research progress and providing opportunities to contribute at all stages. However, the impact on the quality of CEI undertaken remotely or via mixed approaches needs to be investigated.

### Language and literacy barriers

Messaging apps were used for ongoing communication with CEI groups, but participants reported language and literacy barriers. Members of a messaging group may speak different dialects or languages depending on where they are based within the country—or across countries. The reach across borders via virtual platforms poses the challenge of finding a common language. Where face-to-face engagement would usually take place with skilled translators, rapid and accurate translation is not an easy option in live virtual chats and meetings.[31]

### Marginalised communities

Participants reported that a common way to reach community members is via gatekeepers, who are often health workers or community leaders (for example, faith leaders or community elders). Accessing marginalised groups via gatekeepers is an established method for reaching a certain cohort of people based on, for example, location or faith.[32] However, participants noted that there can be limitations when accessing these communities via gatekeepers as they may restrict access to certain groups or individuals.

Another factor is the influence and power that gatekeepers often have over their communities. Some gatekeepers may be resistant to or sceptical of change and may use their influence over the community to sway their opinions.[33] [34] Restrictions on face-to-face meetings mean that this issue may be enhanced, due to researchers relying fully on gatekeepers communicating potentially complex studies to communities—potentially influencing audiences with biased views.

### Limitations

The data for this research have come from a relatively small sample and were collected soon after COVID-19 had become a global pandemic, and so provide only a snapshot of the CEI activities being done in global health research. However, the authors saw this as an opportunity for shared learning in a unique situation and recommend that further research is done looking at the long-term impact of the COVID-19 pandemic on CEI activities.

Also in this study, the authors did not specifically invite community members to participate in the survey or round table. Given the resource constraints (mainly time-sensitivity and no allocated budget) and the potential barriers to inviting community members who are part of the research team (ie, language, training, equal representation from all identified LMICs), this undertaking was not feasible for this piece of research. Participation of community members would have added a different perspective to our research which under the given circumstances was not feasible (as outlined in our paper, reaching out to community members during the pandemic is challenging), but would have also needed more resources, potential funding and manpower, which the authors did not have access to given that this was and is additional research alongside our normal workload. The authors recommend a community member perspective on the impact of the COVID-19 pandemic on CEI activities be explored further.

### CONCLUSION

CEI adaptations were made in a fast-changing environment. There is much learning to be had for future similar localised and global situations, as well as during the ongoing COVID-19 pandemic. As countries gradually move towards a 'new normal', the way CEI is done may have changed forever as many researchers have experienced benefits from their adapted CEI activities. As research continues in uncertain pandemic (or communicable disease) situations, we recommend that research teams and funders should consider: using time effectively to plan for CEI activities when it is not currently possible; sharing relevant networks and resources; promoting acceptance and adoption of online technologies; adapting to the changing needs of the community and developing contingency CEI plans to tackle the unexpected.

**Acknowledgements** We thank the UK National Institute for Health Research (NIHR) and Department of Health and Social Care staff who provided input into the paper based on their personal views. These included Gary Hickey (senior research manager, community engagement and involvement, UK NIHR), Aaronjay Tidball (global health research programme officer, science, research and evidence directorate, DHSC), Charlotte Seeley-Musgrave (global health research officer,

global health research partnerships, DHSC), Patrick Wilson (head of global health communications and stakeholder engagement, UK NIHR), Sarah Puddicombe (assistant director, global health research, UK NIHR) and Jon Cole (assistant director, public involvement and external engagement, UK NIHR). We are also grateful to the NIHR CEI Global Health Advisory Network members who participated in the round table discussion.

**Contributors** KK, HS, KP and DT contributed equally to the design and implementation of the research, to the analysis of the results and to the writing of the manuscript. KK is the guarantor for the paper and accepts full responsibility for the finished work and/or the conduct of the study, had access to the data, and controlled the decision to publish.

**Funding** This research was funded internally by the National Institute of Health Research (NIHR). Award/Grant number is not applicable.

**Competing interests** None declared.

**Patient consent for publication** Not required.

**Ethics approval** We obtained full ethical approval for the research from the University of Southampton's Faculty of Medicine Ethics Committee (no. 53873).

**Provenance and peer review** Not commissioned; externally peer reviewed.

**Data availability statement** All data relevant to the study are included in the article or uploaded as supplemental information. Additional data are available upon request.

**ORCID iD**
Katie Porter http://orcid.org/0000-0002-7841-2588

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
