## [Reviewer comments · BMJ Open]

ARTICLE DETAILS

TITLE (PROVISIONAL)	Challenges and solutions: Surveying researchers on what type of community engagement and involvement activities are feasible in low and middle income countries during the COVID-19 pandemic
AUTHORS	Kroese, Karolin; Porter, Katie; Surridge, Heidi; Tembo, Doreen

VERSION 1 – REVIEW

REVIEWER	Mukherjee, Mrinal The West Bengal University of Teachers' Training, Education Planning and Administration, Department of Teacher Education
REVIEW RETURNED	09-Jun-2021

GENERAL COMMENTS	Objectives of the study is extremely relevant in the Time of pandemic. The title may be reconsidered as the term 'during COVID-19' is inconclusive. The term 'pandemic' need to include in title. How the respondents are selected for Round Table discussion need to mention specifically. Furthermore, design of such discussion need to clarify further like how the discussion was initiated by lead researcher and who played the role of facilitators and how. How the researchers ensure the trust and comfortless of the respondent participant and created conducive climate for them, that also need to justify.
--

REVIEWER	Yaphe, Sean Henry Ford Hospital, Family Medicine
REVIEW RETURNED	21-Jun-2021

GENERAL COMMENTS	This is an important manuscript that acknowledges the challenges and provides solutions to rectify the effects of the COVID-19 pandemic on community-based research in low and middle income communities. It establishes concrete solutions to help move research along and improve outcomes in these communities. 1) Please specify some qualitative outcome measures that were intended with the initial creation of this study 2) Please clarify why community partners (patient and public involvement) were not included in the survey and round table discussions. These community partners are researchers in these initiatives and have an important voice to help establish challenges and provide solutions so that research can continue.
---

REVIEWER	Kalbarczyk, Anna Johns Hopkins University Bloomberg School of Public Health, International Health
-----------------	--

REVIEW RETURNED	28-Jun-2021
-------------

GENERAL COMMENTS	Thank you for providing the opportunity to review this paper. Community engagement is a vital part of global health research, and it's something we need to strengthen across disciplines, particularly as we explore routes to actively decolonize global health and our approaches to global health research. I was surprised to see no authors from LMIC institutions on this list. I think this is a major flaw of the reporting, particularly when you're talking about community engagement IN LMICs. The authors argue in the Background that CEI can build on local capacity and empower people at grassroots level. To me this starts by conducting research and publishing alongside researchers based in LMICs. I also imagine that during lockdown, authors based in LMICs could provide much needed perspectives on many of the issues raised here regarding conducting research and deriving recommendations from the survey and roundtable. Round table - I would like more information about the round table participants. How were they selected? Results - approximately how large was the likely sample, beyond the NIHR funded awardees? I'd like to know more about the other respondents, not just the professors and managers based at UK universities. Were there other respondents actually based in LMIC institutions? I didn't see a demographics table included. Did you find that projects/research moved forward even without CEI? That is, did it become deprioritized as an approach when it became more difficult to conduct? If this emerged in the survey or roundtables, it's a major area for further reflection. The authors mention PPE/facemasks in the results and describe that community may have exhibited caution or mistrust when researchers weren't easily identifiable. Considering the lack of availability of PPE in many settings, this caution/mistrust could also be due to that lack of access- that researchers have PPE but community members do not? The section in inclusivity and access to technology is strong! The discussion section largely repeats the key findings from the results without providing further insight into CEI and pandemic lockdown. This feels like a missed opportunity for the authors to make important connections. I suggest looking at some of the resources from the Research Fairness Initiative about inclusive partnerships, how to measure, and develop them, and what this might mean for the findings about developing new partnerships vs. maintaining established ones during pandemic? I recommend the authors reformat to focus on their recommendations throughout the discussion section rather than in the conclusion.
---

VERSION 1 – AUTHOR RESPONSE

Reviewer: 1

1) Objectives of the study is extremely relevant in the Time of pandemic. The title may be

reconsidered as the term 'during COVID-19' is inconclusive. The term 'pandemic' need to include in title.

Response: The term "the COVID-19 pandemic" has been used in the title rather than "COVID-19".

2) How the respondents are selected for Round Table discussion need to mention specifically.

Response: Participants of the round table were all members of the NIHR Global Health CEI Advisory Network, which is a group of academics, researchers, and consultants with knowledge and experience of CEI in LMICs. Members of the group were initially invited to join by the NIHR CEI leadership team to share knowledge and experience around CEI and to provide advice on NIHR's CEI strategy and operation. The Network members include representation from charities and universities based both in the Global North and the Global South. All members share an interest in CEI but other research interests range from epidemiology to mental health to social anthropology. More information can be found on the following webpage: <https://mesh.tghn.org/programme-hubs/nih-global-health-cei-involvement/>. Further reasoning about why they were chosen for the round table is given under "Methods".

3) Furthermore, design of such discussion need to clarify further like how the discussion was initiated by lead researcher and who played the role of facilitators and how.

Response: Two of the authors (Karolin Kroese and Heidi Surridge) facilitated the online round table (via Zoom). The discussion was guided by a series of questions agreed by the research team prior to the round table (appendix 2).

4) How the researchers ensure the trust and comfortless of the respondent participant and created conducive climate for them, that also need to justify.

Response: A participant information sheet was provided to potential survey participants (either via email or on the webpage) explaining the purpose of the survey and ensuring their anonymity in the final manuscript.

All participants had completed consent forms and received an information sheet prior to the round table. Three of the authors (Katie Porter, Heidi Surridge, and Doreen Tembo) work regularly with this Network to help shape and inform NIHR's CEI portfolio, and so relationships with participants were already established.

Reviewer: 2

1) Please specify some qualitative outcome measures that were intended with the initial creation of this study.

Response: The intended outcome measures were the number of CEI activities halted, adapted and continued as planned because of the COVID-19 pandemic; where CEI is possible; how it has been adapted; what the challenges and successes were; and the potential impact of adapted or halted CEI on global health research. These have been reported on in the manuscript. These outcome measures have been clarified in the manuscript.

2) Please clarify why community partners (patient and public involvement) were not included in the survey and round table discussions. These community partners are researchers in these initiatives and have an important voice to help establish challenges and provide solutions so that research can continue.

Response: Global health research teams undertaking research in LMICs were invited to fill in the survey. The invite was sent to the recorded project lead, however, they were asked to consult their wider team. This included any community members, where applicable and relevant. Community members were not invited to the round table, as members were part of the NIHR Global Health CEI Advisory Network. Given the resource constraints (mainly time-sensitivity and no allocated budget) and the potential barriers to inviting community members who are part of the research team (i.e., language, training, equal representation from all identified LMICs), this undertaking was not feasible.

for this piece of research. This information has been added to the “Limitations” section of the manuscript.

Reviewer: 3

1) I was surprised to see no authors from LMIC institutions on this list. I think this is a major flaw of the reporting, particularly when you're talking about community engagement IN LMICs. The authors argue in the Background that CEI can build on local capacity and empower people at grassroots level. To me this starts by conducting research and publishing alongside researchers based in LMICs. I also imagine that during lockdown, authors based in LMICs could provide much needed perspectives on many of the issues raised here regarding conducting research and deriving recommendations from the survey and roundtable.

Response: This was a reactive piece of monitoring and evaluative work initiated by NIHR and an initial query raised by a global health CEI specialist (Karolin Kroese) with a primary purpose of understanding the impact of the pandemic on the research community working in LMICs.

Some of the survey and round table participants were based in LMICs (as shown in the new demographics table (table 1). As part of the planning process, representative individuals from LMIC-based research teams, including members of the public, and CEI professionals were asked to review the survey questions to ensure relevant information on CEI activities is being captured. They were also asked to trial the online survey tool for usability and clarity prior to it being circulated. We also consulted the CEI leadership team at the NIHR and DHSC in reviewing the outcomes of the round table and wider research before writing this up. In our opinion, we therefore did involve the right people in the design and analysis of our research. The patient or public would have added a different angle/perspective to our research which under the given circumstances was not feasible (as outlined in our paper, reaching out to community members during the pandemic is challenging), but would have also needed more resources, potential funding and manpower, which we did not have access to given that this was and is an ad hoc bit of research alongside our normal workload. The patient angle could be a follow-on piece.

Two of the authors (Karolin Kroese and Doreen Tembo) are also (or have been recently) part of funded Global Health Research teams and have led CEI activities based in LMICs, so their in-country experience has been invaluable to the development of this research

2) Round table - I would like more information about the round table participants. How were they selected?

Response: Participants of the round table were all members of the NIHR Global Health CEI Advisory Network, which is a group of academics, researchers, and consultants with knowledge and experience of CEI in LMICs. Members of the group were initially invited to join by the NIHR CEI leadership team to share knowledge and experience around CEI and to provide advice on NIHR's CEI strategy and operation. The Network members include representation from charities and universities based both in the Global North and the Global South. All members share an interest in CEI but other research interests range from epidemiology to mental health to social anthropology. More information can be found on the following webpage: <https://mesh.tghn.org/programme-hubs/nih-global-health-cei-involvement/>

3) Results - approximately how large was the likely sample, beyond the NIHR funded awardees?

Response: As we used a snowballing technique, as well as purposive sampling techniques, we do not know how many people who are not NIHR-funded awardees received the survey. The survey was posted on the Mesh Community Engagement Network website which is open access so we do not have data on how many people accessed it via this route.

In terms of the sample who participated in the survey, 31 surveys were returned, 27 of which were in receipt of NIHR funding so 4 surveys were returned from non-NIHR funded research teams.

4) I'd like to know more about the other respondents, not just the professors and managers based at UK universities. Were there other respondents actually based in LMIC institutions? I didn't see a demographics table included.

Response: Over half of survey and round table participants were based in the UK, but some had experience undertaking CEI activities in LMICs and some consulted with LMIC-based members of their research team before summarising their teams' responses in the survey. There were 8 (24%) survey participants and 7 (32%) round table participants based in LMICs. A demographics table showing the location and role of participants has been added to the manuscript (table 1).

5) Did you find that projects/research moved forward even without CEI? That is, did it become deprioritized as an approach when it became more difficult to conduct? If this emerged in the survey or roundtables, it's a major area for further reflection.

Response: Some survey participants reported that their planned CEI activities were not feasible in the pandemic situation so these activities were paused and the teams continued with research activities that were feasible. It must be considered that where CEI activities were paused, participants reported that they were in the process of adapting their CEI and research activities to adhere to pandemic restrictions (e.g. switch in remote, online methods). We have added better reflection on this point in the Discussion.

6) The authors mention PPE/facemasks in the results and describe that community may have exhibited caution or mistrust when researchers weren't easily identifiable. Considering the lack of availability of PPE in many settings, this caution/mistrust could also be due to that lack of access- that researchers have PPE but community members do not?

Response: This is a good point that we did not initially consider. We have added a short reflection on this point in the Discussion.

7) The discussion section largely repeats the key findings from the results without providing further insight into CEI and pandemic lockdown. This feels like a missed opportunity for the authors to make important connections. I suggest looking at some of the resources from the Research Fairness Initiative about inclusive partnerships, how to measure, and develop them, and what this might mean for the findings about developing new partnerships vs. maintaining established ones during pandemic? I recommend the authors reformat to focus on their recommendations throughout the discussion section rather than in the conclusion.

Response: We have reformatted the manuscript so that the recommendations into the Discussion as suggested and removed any areas of repetition. Some key points from the findings are repeated in the Discussion, but this is with the intent to reflect on these points from the perspective of the author and wider literature.

VERSION 2 – REVIEW

REVIEWER	Kalbarczyk, Anna Johns Hopkins University Bloomberg School of Public Health, International Health
REVIEW RETURNED	05-Sep-2021
GENERAL COMMENTS	Thank you for addressing my previous comments. The discussion is much more compelling with the redesign.